# From Quiescence to Activation: The Reciprocal Regulation of Ras and Rho Signaling in Hepatic Stellate Cells

**DOI:** 10.3390/cells14090674

**Published:** 2025-05-05

**Authors:** Saeideh Nakhaei-Rad, Silke Pudewell, Amin Mirzaiebadizi, Kazem Nouri, Doreen Reichert, Claus Kordes, Dieter Häussinger, Mohammad Reza Ahmadian

**Affiliations:** 1Institute of Biochemistry and Molecular Biology II, Medical Faculty and University Hospital Düsseldorf, Heinrich Heine University Düsseldorf, 40225 Düsseldorf, Germany; silke.pudewell@hhu.de (S.P.); amin.mirzaiebadizi@hhu.de (A.M.); kazem.nouri@vch.ca (K.N.); 2Stem Cell Biology and Regenerative Medicine Research Group, Institute of Biotechnology, Ferdowsi University, Mashhad 9177948974, Iran; 3Department of Pathology and Laboratory Medicine, University of British Columbia, Vancouver, BC V6T 2B5, Canada; 4IUF—Leibniz Research Institute for Environmental Medicine, 40225 Düsseldorf, Germany; doreen.reichert@iuf-duesseldorf.de; 5Institute of Stem Cell Research and Regenerative Medicine, Medical Faculty, University Hospital Düsseldorf, Heinrich Heine University Düsseldorf, 40225 Düsseldorf, Germany; claus.kordes@hhu.de; 6Heinrich Heine University Düsseldorf, 40225 Düsseldorf, Germany; haeussin@uni-duesseldorf.de

**Keywords:** chronic liver disease, cirrhosis, hepatic stellate cells, liver fibrosis, quiescent state, Ras GTPases, Rho GTPases, small molecule inhibitors

## Abstract

Chronic liver diseases are marked by persistent inflammation and can evolve into liver fibrosis, cirrhosis, and hepatocellular carcinoma. In an affected liver, hepatic stellate cells (HSCs) transition from a quiescent to an activated state and adopt a myofibroblast-like cell phenotype. While these activated cells play a role in supporting liver regeneration, they can also have detrimental effects on liver function as the disease progresses to fibrosis and cirrhosis. These findings highlight the dynamic switching between different signaling pathways involving Ras, Rho GTPases, and Notch signaling. Notably, two specific members of the Ras and Rho GTPases, Eras and Rnd3, are predominantly expressed in quiescent HSCs, while Mras and Rhoc are more abundant in their activated forms. In addition, this study highlights the critical role of cytosolic Notch1 in quiescent HSCs and Rock in activated HSCs. We hypothesize that distinct yet interdependent intracellular signaling networks regulate HSC fate decisions in two key ways: by maintaining HSC quiescence and homeostasis and by facilitating HSC activation, thereby influencing processes such as proliferation, transdifferentiation, and mesenchymal transition. The proposed signaling model, combined with specific methodological tools for maintaining HSCs in a quiescent state, will deepen our understanding of the mechanisms underlying chronic liver disease and may also pave the way for innovative therapies. These therapies could include small molecule drugs targeting Ras- and Rho-dependent pathways.

## 1. Introduction

Chronic liver diseases have various underlying causes that lead to persistent liver inflammation, which can result in liver fibrosis, cirrhosis, and hepatocellular carcinoma. These are responsible for more than 1 million deaths worldwide each year [1]. At the onset of fibrosis in liver disease, hepatic stellate cells (HSCs; also called Ito cells, lipocytes, fat-storing cells, or perisinusoidal cells) activate and transdifferentiate into contractile, matrix-producing myofibroblast-like cells, which are central events in hepatic fibrogenesis [2]. Myofibroblast-like cells produce the fibrous scar in hepatic fibrosis. HSCs contribute to 5–8% of total liver-resident cells and are located as pericytes of sinusoidal endothelial cells in the space of Disse bordered by hepatocytes [3]. HSCs play pivotal roles in liver development, immunomodulation, regeneration, and pathology [4]. They exhibit remarkable plasticity in their phenotype, gene expression profile, and cellular functions. In a healthy liver, HSCs remain quiescent and store vitamin A mainly as retinyl palmitate in cytoplasmic membrane-coated vesicles. Moreover, HSCs typically express ectodermal and mesodermal markers, i.e., glial fibrillary acidic protein (Gfap) and Desmin. They possess characteristics of stem cells, like the expression of *Wnt*, *Notch*, and *Eras* (embryonic stem cell-expressed Ras), which are required for developmental fate decisions [4,5]. HSCs have an expression profile that is highly reminiscent of mesenchymal stem cells. Because of their typical functions, such as differentiation into adipocytes and osteocytes and the support of hematopoietic stem cells, HSCs have been defined as liver-resident mesenchymal stem cells [6,7]. In recent years, it has become increasingly clear that pericytes support the vascular system by having a stem cell character [8], and they are also involved in developing chronic diseases such as fibrosis [9].

During activation, HSCs release vitamin A, upregulate various genes and differentiation markers, including α-smooth muscle actin (α-Sma) and collagen type I, and downregulate quiescence markers such as Gfap. Physiologically, HSCs represent pericytes and can produce an extracellular matrix (ECM). In pathophysiological conditions, the sustained activation of HSCs causes the accumulation of ECM in the liver. Fibrosis is a dynamic process involving cross-talk between HSCs, sinusoidal endothelial cells, liver-resident and -infiltrating immune cells, and hepatocytes [10]. Therefore, it is worthwhile to reconsider the impact of different signaling pathways on HSC fate decisions to modulate them. For example, quiescent HSCs might contribute to maintaining liver tissue but not fibrosis. Activated HSCs are multipotent cells, and studies have revealed a new aspect of HSC plasticity, i.e., their differentiation into liver progenitor cells during liver regeneration [11]. To date, several growth factors like platelet-derived growth factor (Pdgf), transforming growth factor-β (Tgfβ), and insulin-like growth factor 1 (IGF1), as well as different signaling pathways, have been described to control HSC activation through effector pathways, including canonical Wnt, Hedgehog, Notch, Ras-Mapk (mitogen-activated protein kinase), Pi3k (phosphoinositide 3-kinase)-Akt-Pkb (protein kinase B), Hippo-Yap (yes-associated protein), and Rho-Rock (Rho-kinase) [5,12,13,14,15,16,17,18]. However, it is necessary to identify the key players controlling HSC activation and to understand the mechanisms controlling HSC fate.

Altered Ras- and Rho-mediated signaling pathways represent some of the earliest events in HSC activation as identified by the present study and act as central hubs for intracellular signaling networks [5,19,20]. Ras and Rho proteins are small GTPases involved in diverse cellular processes, including intracellular metabolism, proliferation, morphogenesis, migration, and differentiation, and thus play critical roles in embryogenesis, development, and tissue remodeling [21]. In our molecular characterization of quiescent and activated HSCs, we identified the reciprocal and paralog-specific expression and activity of Ras- and Rho-related GTPases and associated signaling components such as Notch, Yap, and Erk [5,22]. Our data revealed a greater abundance of Eras, Rnd3, and Notch1 in the quiescent state, correlating with heightened activity in the Eras-Pi3k-Akt and Notch-Rnd3 pathways. Conversely, elevated levels of Mras and Rhoc were associated with activated HSCs, reflecting increased activities in the Ras-Mapk and Rhoc-Rock axes. These findings provide new insights into the regulatory mechanisms distinguishing quiescent from activated HSCs and highlight the key molecular switches controlling this process.

## 2. Materials and Methods

The cell isolation procedure was approved by the local authority for animal protection (Landesamt für Natur, Umwelt und Verbraucherschutz Nordrhein-Westfalen, LANUV, Recklinghausen, Germany; reference number 84-02.04.2015.A287). Male Wistar rats (500–600 g) were obtained from the local animal facility of Heinrich Heine University (Düsseldorf, Germany) and used for the isolation of HSCs as described previously [5]. The number of animals used per experimental set was usually one, which resulted in a total number of animals of 3 to 5.

Cell isolation and culture, quantitative polymerase chain reaction (qPCR), immunoblotting, and confocal imaging were carried out according to previously described protocols [5]. Briefly, HSCs were seeded on stiff plastic shortly after their isolation and maintained as a monoculture in the presence of fetal calf serum (FCS). The combination of the rigid culture surface and the factors present in FCS, along with the conditions of monoculture, facilitates the activation of HSCs. Activation begins soon after cell isolation, with noticeable changes occurring between the 2nd and 3rd days of culture [1,2,3]. The qPCR primer sequences are listed in Appendix A. The antibodies used for immunoblotting and confocal imaging are listed in Appendix A. CRISPR/Cas9 genome editing was performed using four different guide RNAs (gRNA; Appendix A) for rat arginase 1 (Arg1; ID 29221) that were designed by using the Chopchop V3 online software tool [23]. The gRNAs were synthesized with the GeneArt Precision gRNA Synthesis Kit (Invitrogen, Carlsbad, CA, USA), pre-incubated with the TrueCut Cas9 Protein v2 (Invitrogen), and finally transferred into freshly isolated HSCs using the 4D-Nucleofector (program DS167; P3 Primary Cell 4D-Nucleofector X Kit L; Lonza, Basel, Switzerland). Control (mock) cells were treated similarly but without the gRNA. The HSCs were then cultured on plastic in DMEM with 10% FCS and 1% penicillin/streptomycin (37 °C, saturated humidity) for three days. The Arg1 KO efficacy was determined with Western blot.

## 3. Results and Discussion

### 3.1. Reciprocal, Paralog-Specific Ras and Rho GTPases in HSC Fate Determination

HSCs are the primary source of myofibroblast-like cells in liver fibrosis and primary liver cancer [24]. Freshly isolated quiescent Gfap-positive HSCs spontaneously undergo activation and transdifferentiation into myofibroblast-like cells when cultured on stiff plastic dishes, as indicated by the presence of α-Sma. We used the plasticity of freshly isolated HSCs as an in vitro cell model to investigate the control mechanisms that trigger their activation. In this regard, we focused on the molecular switches of Ras GTPases and their expression changes. The mRNA analysis revealed the remarkable changes in the family members of Ras between two states, quiescent vs. activated HSCs [5].

These data indicated the upregulation of a few genes related to the Ras GTPases, such as *Mras* (muscle Ras oncogene homolog), during the HSC activation phase. Eras, a unique member of the Ras family, was first identified in undifferentiated embryonic stem cells [25]. Unlike other members of the Ras family, Eras is not ubiquitously expressed, and its expression is cell-type and tissue-specific. We exclusively found Eras within quiescent but not activated HSCs (Figure 1A,B) [5]. In addition, Eras also regulates the Hippo pathway (Rassf-Mst-Lats), leading to the phosphorylation and subsequent degradation of Yap [5].

Mapk and Pi3k-Akt are known as primary Ras-dependent signaling pathways. The activity of the Mapk signaling pathway is critical for cell proliferation and differentiation. It is significantly higher in activated than in quiescent HSCs, as evidenced by increased levels of phosphorylated Erk (p-Erk1/2) (Figure 1B). In contrast, the activation of Akt via the Pi3k and mTor complex 2 (mTorc2) pathways is particularly important in quiescent HSCs. Notably, Akt is more extensively phosphorylated at serine 473 by the rapamycin-insensitive mTorc2 complex compared with threonine 308 via the Pi3k-Pdk1 pathway, affecting various cellular processes in HSCs [5].

Remarkably, the downregulation of Eras was found to correlate with the upregulation of Mras (Figure 1B). We have previously demonstrated that Eras preferentially binds to the Pi3k effector protein and activates the Pi3k-Pdk1-Akt axis, not the Mapk pathway [5]. This suggests that Eras signaling through the Pi3k-Pdk1 and mTorc2 pathways fully activates Akt at both phosphorylation sites, thereby maintaining HSCs in their quiescent state. The increased activation of the Mapk pathway in activated HSCs is consistent with the increased expression of Mras, which has been shown to form a phosphatase holoenzyme complex with Shoc2 and PP1, controlling Raf1 dephosphorylation at S259 and initiating Mapk activation.

In addition to Ras family members, we observed the reciprocal expression of several Rho GTPases, including Rnd3 (Rho Family GTPase 3), and Rhoc (Ras Homolog Family Member C) (Figure 1A). Rho family proteins are key regulators of the actin cytoskeleton and influence various cellular processes, morphogenesis, proliferation, migration, and differentiation, which are fundamental for HSC activation [21]. Initial quantitative mRNA analyses of Rho family GTPases revealed the remarkable upregulation of Rnd3 in quiescent HSCs and Rhoc in activated HSCs (Figure 1A). Immunoblot analysis using validated antibodies against the Rho paralogs confirmed that Rnd3 and Rhoc are reciprocally expressed in quiescent and activated HSCs (Figure 1B). The high levels of Rhoc appear critical for HSC activation, but its activation might also be tightly regulated, most likely through its negative regulator p190RhoGAP (p190Rho guanosine triphosphatase-activating protein or p190GAP) [26]. Furthermore, it is striking that Rhoc and p190GAP are largely expressed in activated HSCs (Figure 1B). The regulation of p190GAP activity has been reported to control mechanical and Tgf-β signaling and, thus, the fibrotic phenotype of idiopathic pulmonary fibroblasts [27]. Therefore, we propose that Rhoc and its inactivator p190GAP may be important components in the regulation of Rock (see below) during HSC activation and myofibroblast formation. Rnd3 has been reported to be a transcriptional target of activated Notch1 as it is more expressed in quiescent HCSs [28].

### 3.2. The Key Role of Cytosolic Notch1 and HSC Quiescence State

Notch proteins (Notch1–4) are conserved transmembrane receptors involved in numerous developmental processes, including stem cell self-renewal. Notch1 is expressed as a target gene of the canonical Wnt signaling pathway in quiescent HSCs [29]. Several studies suggest that Notch1 signaling regulates stem cell maintenance and counteracts their differentiation [30,31,32]. Quantitative mRNA analysis revealed the reciprocal expression of Notch1 and Notch2 during activation. In contrast to *Notch2*, *Notch1* and its target gene *Hes1* were found to be predominantly expressed in quiescent HSCs (Figure 2A).

The canonical Notch1 signaling pathway involves the sequential proteolytic processing of Notch1 by Adam17 (a disintegrin and metalloprotease 17) and γ-secretase to release the intracellular domain of Notch1 (N1ICD), which, in turn, translocates to the nucleus and, as a multimeric protein complex, displaces transcriptional co-repressors by recruiting transcriptional co-activators. N1ICD has been shown to control transcription and maintain the undifferentiated status of stem cells [33,34]. As shown in Figure 2B, Notch1 is present in quiescent HSCs largely in a processed form as 120 kDa N1ICD, whereas Yap is exclusively present in activated HSCs [5], consistent with the expression of its target gene *Notch2* (Figure 2A). Subsequent confocal imaging revealed that Notch1 and/or N1ICD are largely present in the cytosol of quiescent HSCs (Figure 2C). Based on these data, we hypothesize that Notch1 activity appears to maintain HSC quiescence downstream of the Eras-Pi3k-Akt pathway; a direct role of Eras in this process remains to be investigated. Although Notch1 and N1ICD were mainly detected in cytosol, we could not rule out the transcriptional activity of Notch1, while the expression of the Notch1 target gene, e.g., *Hes1* in quiescent HSCs (Figure 2A), indicates that N1ICD is still transcriptionally active. Interestingly, Hes1 has been shown to negatively regulate α-Sma and Col1α2 [35]. Thus, N1ICD-Hes1 activity may provide an anti-fibrotic strategy.

These results suggest that HSCs use different pathways downstream of Eras to determine their fate. In quiescent HSCs, the activation or differential regulation of the Pi3k-Pdk1 and *mTorc2* pathways by Eras appears essential for HSC quiescence [5]. In contrast, in activated HSCs, signaling transitions from Eras to Mras and from Rnd3 to Rhoc, activating the Raf-Mek-Erk and Rock pathways while inactivating the Hippo pathway. This shift facilitates cellular proliferation and mesenchymal transition, ultimately driving HSCs to transdifferentiate into myofibroblast-like cells, a critical event in liver fibrosis.

Therefore, we propose a hypothetical model in which Eras-Pi3k-mediated Akt activation counteracts the activation of the Mras-Raf1-Mek-Erk and Rhoc-Rock-Gfap axes (Figure 3) [36,37]. This can be achieved by expressing the Notch1 target gene Rnd3 (Figure 2A) [28], a Rhoc antagonist that directly binds and inhibits Rock [38].

### 3.3. Rock a Key Determinant of HSC Activation

In liver fibrosis, the activation of HSCs involves phenotypic transformation into profibrotic and myofibroblastic cells with the increased contraction and secretion of ECM proteins [2,39]. In this context, the Rhoc-Rock signaling pathway may play a key role in orchestrating cytoskeletal reorganization and mobility via the non-receptor tyrosine kinase Src, thus playing a critical role in HSC activation and hepatic fibrogenesis [40]. To investigate this issue, we treated freshly isolated HSCs with the Rock inhibitor Y-27632 and analyzed HSC morphology using confocal imaging. Figure 2D clearly shows the delayed activation of HSCs in the presence of Y-27632; the cells were still Gfap-positive, and the stellate-like morphology at d4 of the culture was very similar to quiescent cells. The activation of the Rhoc-Rock pathway in HSCs can control the phosphorylation of Gfap filaments and their subsequent disassembly [41] and inhibit Hippo pathways leading to Yap activation [42]. This suggests that the activation of the Rhoc-Rock pathway is one of the early processes during HSC activation and, ultimately, pathogenesis, which is antagonized by the Notch1-induced expression of Rnd3 in quiescent HSCs (Figure 2A).

### 3.4. Role of Eras-Arg1-Polyamine Axis in HSC Homeostasis

Two important processes of HSCs—self-renewal and autophagy—are important prerequisites for the maintenance of liver homeostasis. These processes depend, among other things, on the availability of polyamines, such as spermidine and spermine which are natural molecules that have a variety of functions, including cell growth, cell differentiation, and cell survival. Studies have shown that spermidine exerts a protective effect on the liver, especially in the context of age-related changes [43]. Spermidine ingestion in mice leads to improvement in liver function by inhibiting HSC activation and liver fibrosis [44].

We recently identified and characterized the Eras-interacting proteins, including Arg1 (arginase 1) [22], a key enzyme of the urea cycle involved in de novo polyamine synthesis. Arg1, which is inversely regulated to iNos (inducible nitric-oxide synthase), is co-expressed and colocalized with Eras at the membrane of freshly isolated HSCs [22]. Arg1 catalyzes the hydrolysis of L-arginine to L-ornithine, which is further converted into polyamines such as putrescine, spermidine, and spermine by Odc1 (ornithine decarboxylase 1), Sds (spermidine synthase), and Sms (spermine synthase), respectively (Figure 3). Notably, polyamines are implicated in a broad range of cellular processes, including cell metabolism, transcription, translation, post-translational modifications (i.e., hypusination), and autophagy [44]. Using specific pharmacological inhibitors of L-arginine metabolism (Figure 3), we found that the inhibition of Arg1 or Odc1 accelerated HSC activation, as indicated by the loss of the stellate-like morphology and lipid droplet storage [22]. These data suggest that Arg1 catalytic activity may be a critical determinant of developmental fate decisions. This is accomplished by two possible polyamine effector pathways: autophagy and self-renewal (Figure 3).

Autophagy is achieved by the synthesis of hypusine from spermidine and the post-translational hypusination of Eif5a (eukaryotic translation initiation factor 5A) [45], which is, in turn, necessary for the translation of Atg3 (autophagy-related protein 3, which is part of the complex for Lc3 (microtubule-associated proteins 1A/1B light chain 3)) lipidation and converts Lc3-I to Lc3-II, therefore being central for autophagosome assembly [46]. Another effector of polyamines is Mindy1, a deubiquitinase that maintains stemness by sustaining Oct4 protein levels and inducing self-renewal in ESCs [47]. The stem cell marker Oct4 is expressed in quiescent HSCs [48]. HSCs are considered liver-resident mesenchymal stem cells, which can differentiate into diverse cell types in response to liver damage [49]. Moreover, the regulation of Mindy1 by the L-ornithine derivative spermine is, like Eras, anchored to the membrane by C-terminal isoprenylation [25,47] and appears to be important for maintaining stem cell properties and controlling the quiescence of HSCs. However, Mindy1 may be one of the many polyamine effectors that participate in the maintenance of HSC quiescence. The role of polyamines in the modulation of autophagy and the quiescence of HSCs can be investigated through Arg1 knockout by pharmacological inhibitors of Arg1 and Odc1 (Figure 3 and Figure 4).

An essential aspect of studying primary HSCs as a model for liver fibrosis is to identify an optimal time window for manipulating freshly isolated cells through gene transfer or knockout while they remain in their quiescent state. Utilizing a recently published protocol, we have successfully established tools and approaches to maintain HSCs in their quiescent state (Figure 4). Cells are transfected on day 0, immediately after isolation, using medium 2. In this medium, HSCs remain undifferentiated for several days. These culture conditions allow for the genetic manipulation of quiescent HSCs, including gene knockouts, such as Arg1 (Figure 4), using CRISPR/Cas9 genome editing. Electroporation-based gene transfer has proven to be the most effective method for the efficient transfection of freshly isolated HSCs. Finally, the medium-preserving quiescence is replaced with medium 1 to facilitate HSC activation and transdifferentiation.

## 4. Conclusions and Future Directions

There is a long list of in vivo and clinical studies on the role of HSCs in the progression of liver diseases, including fibrosis, cirrhosis, and cancer. With the onset of fibrosis in liver disease, HSCs activate and transdifferentiate into myofibroblast-like cells that produce large amounts of collagen and other extracellular matrix components. However, ethical concerns, the use of animals in drug safety studies, the timing of experiments, and costs led to the consideration of replacing animal fibrosis models with in vitro HSC culture systems [50]. Furthermore, in vitro experiments allow us to study the complex mechanisms involved in HSC activation and fibrosis development, to test cell-targeted stimulation or genetic manipulation (CRISPR/Cas9 knockout, overexpression, or treatment using specific pharmacological inhibitors), and to discover new anti-fibrogenic targets (Figure 3). As an in vitro model system for liver fibrosis, freshly isolated rat HSCs spontaneously activate into myofibroblast-like cells when cultured on stiff plastic dishes (Figure 4; see also [51,52]). Elevated mechanical stimuli are known to promote Yap transcriptional activity [53] and trigger liver fibrosis in vivo [54].

Since liver fibrosis begins as an intrinsic signaling process and, in its later stages, becomes dependent on microenvironmental factors—such as impaired blood flow due to inflammatory processes—it is evident that various stimuli induce selective signaling pathways involving Src, Rho, and Rock. This activation triggers the nuclear translocation of Yap, which increases the release of growth factors, such as Ctgf (connective tissue growth factor), by HSCs. In turn, this enhances the activation of the Mapk pathway (Figure 3). We therefore hypothesize that the various signaling proteins and their associated intracellular signaling pathways, on the one hand, maintain the quiescent state of HSCs and thus their homeostasis and, on the other hand, contribute to HSC activation and therefore control processes that are essential for the transition from a quiescent to a proliferative, migratory, and fibrogenic phenotype (i.e., myofibroblast-like cells) [17]. Therefore, targeting pathways involved in HSC activation represents a promising strategy to prevent the development and progression of liver fibrosis. The co-administration of sorafenib, an oral RAF kinase inhibitor, can prevent Erk activation in activated HSCs and has shown anti-fibrotic effects in a CCl_4_-induced murine model [55]. In addition, blocking receptor tyrosine kinases such as Pdgfrβ by crenolanib via drinking water can improve thioacetamide-induced liver fibrosis in rats [56].

## Figures and Tables

**Figure 1 cells-14-00674-f001:**
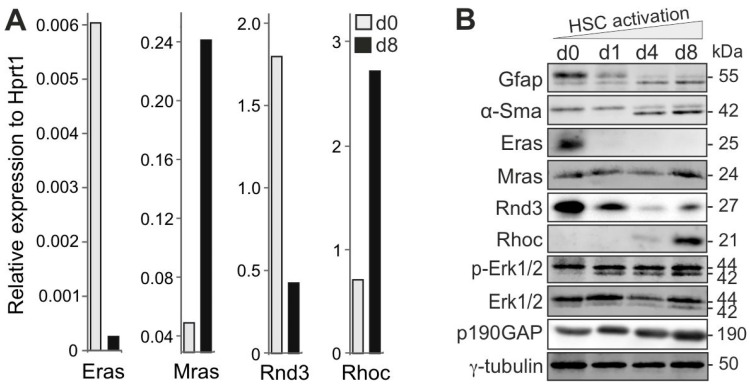
Reciprocal Ras signaling activities in quiescent and activated HSCs. (**A**) Differential *Eras*, *Mras*, *Rnd3*, and *Rhoc* expression in quiescent (d0) versus activated (d8) HSCs was analyzed with qPCR. Hypoxanthine-guanine phosphoribosyl transferase 1 (Hprt1) was used as a normalization control (n = 3). (**B**) Immunoblotting was performed to detect various signaling and marker proteins, as indicated, during HSC activation (n = 2). This analysis provides insight into the changes in protein expression associated with HSC activation. “d” stands for “day”. Please check Appendix A for the original data of the Western blots.

**Figure 2 cells-14-00674-f002:**
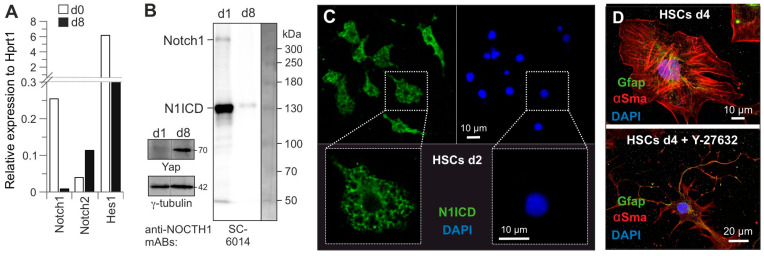
The significance of cytosolic Notch1 in quiescent HSCs and Rock in activated HSCs. (**A**) While *Notch1* and its target gene, *Hes1*, are downregulated during HSC activation, the expression of *Notch2*, a Yap target gene, is strongly upregulated. (**B**) In contrast to Yap, the Notch1 protein is predominantly present in HSCs (d1) as the Notch1 intracellular domain (N1ICD; Cell Signaling SC-6014). Full-length Notch1 was detected at 300 kDa. The Yap and γ-tubulin loading control blots have been trimmed to remove d1 and d4 (dashed line) (n = 2). (**C**) Full-length Notch1 and N1ICD are strikingly localized in the cytosol of HSCs (d2) using a cell signaling antibody (#4380). (**D**) Rock inhibition with 10 µM of Y-27632 (#S1049, Selleckchem) blocked culture-induced HSC activation, most likely by inhibiting the Rock-mediated morphological transition of HSCs. Untreated cells exhibit the cell shape of activated HSCs, which are much larger and contain more α-Sma, whereas Y-27632-treated cells are still stellate cells at d4 with Gfap-containing processes that more closely resemble quiescent HSCs. “d” stands for “day”.

**Figure 3 cells-14-00674-f003:**
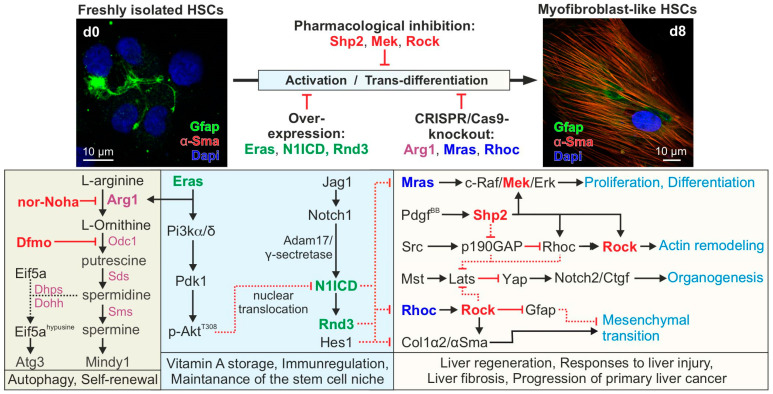
Proposed model of the reciprocal regulation of Ras and Rho GTPase-driven signaling pathways in quiescent vs. activated HSCs. Different signaling networks in quiescent and activated HSCs are subject to reciprocal, paralog-specific regulation by different signaling molecules that are critical for determining developmental fate decisions. L-arginine metabolism to polyamine derivatives, catalyzed by different enzymes, may control processes such as autophagy and self-renewal. The effects of Eras on the Pi3k/Akt axis may contribute to the maintenance of the quiescent state of HSCs by activating Notch1 signaling activity. This, in turn, leads to the inhibition of Mras/Raf1, Rhoc/Rock, and reduced levels of active Yap protein and Yap-mediated signaling, thereby inhibiting proliferation and mesenchymal transition, leading to HSC activation and myofibroblast formation. In addition, the reciprocal expression patterns of Notch1 and Gfap relative to Notch2, Collα2, and α-Sma that result from elevated Yap are another key to this regulatory process. We hypothesize that (i) the overexpression of Eras, N1ICD, or Rnd3 could either maintain HSC quiescence or significantly delay their activation; (ii) the CRISPR/Cas9 knockout of Arg1, Mras, or Rhoc could interfere with HSC activation; and (iii) the pharmacological inhibition of Shp2, Mek, or Rock could block HSC activation ex vivo and prevent the progression of liver fibrosis in vivo. See text for details. NOTE: The proposed model depicted here is hypothetical and based on the evaluated status of selected molecules. “d” stands for “day”.

**Figure 4 cells-14-00674-f004:**
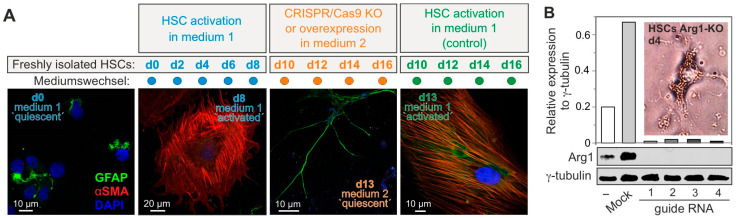
Culture conditions for the long-term maintenance of HSC quiescence. (**A**) Freshly isolated rat HSCs rapidly differentiate into myofibroblast-like cells (α-Sma^high^) when cultured in the Dulbecco’s modified Eagle’s medium (DMEM) supplemented with 15% fetal calf serum and 50 units of penicillin/streptomycin (medium 1). When cultured for 8 days in medium 2, containing DMEM supplemented with 2% FBS, 50 ng/mL of insulin, 10 µM of retinol, 100 µM of oleic acid, 100 µM of palmitic acid, 20 ng/mL of Egf and 10 ng/mL of Fgf2, as shown in this example, they form their characteristic stellate-shaped morphology (Gfap^high^). The replacement of medium 2 with medium 1 led to the activation of the HSCs and their transdifferentiation into the typical myofibroblast-like cells (α-Sma^high^). Gfap is stained green, α-Sma is stained red, and DNA is stained blue with DAPI (n = 2). (**B**) Under the conditions used in (**A**), the Arg1 gene was successfully knocked out in HSCs on day 0 by CRISPR/Cas9 using four different gRNAs and Nucleofector technology (see Methods). Arg1 KO was also verified with Western blot analysis on day 3 of HSC culture in medium 2. cLSM showed a strikingly delayed activation of HSCs (they retained more and larger lipid droplets and appeared star-shaped) (n = 2). “d” stands for “day”.

## Data Availability

It is affirmed that no new data were generated while compiling this manuscript. All referenced data sources are openly accessible and appropriately cited within the manuscript. Please do not hesitate to contact the corresponding author if any additional information or clarification is required.

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
