# Peer review of "From Quiescence to Activation: The Reciprocal Regulation of Ras and Rho Signaling in Hepatic Stellate Cells"

_cells, 2025, doi:10.3390/cells14090674_

Round 1
Reviewer 1 Report
Comments and Suggestions for Authors
The title accurately reflects the content of the paper. The figures and tables are well-designed, and the references are of high quality. The research offers valuable insights into the mechanisms of liver fibrosis. Here are some suggestions I would like to offer:
- The interplay between RAS, RHO GTPase, and other critical signaling proteins in the activation of hepatic stellate cells (HSCs) could be described in more detail in the abstract and introduction.
- Please provide the animal ethics certification, and clearly describe the specific details of animal and cell procedures. The experiments in this paper seem to focus only on phenotypic validation, lacking mechanistic verification, which reduces the persuasiveness of the findings. It is recommended to include additional mechanistic experiments at the cellular level to enhance the credibility of the results. Additionally, a few schematic diagrams illustrating the proposed mechanisms could be included to provide further clarification.
- Please clearly label the specific kDa values of the molecules in the figures. Additionally, the raw data section appears to be incomplete, as the Western blot protein markers and biological replicates are not fully provided. The authors are encouraged to improve and complete this section.
- Please ensure that the dilution ratios of all antibodies are clearly and thoroughly specified by the authors.
- Please clearly specify the number of base pairs in the PCR primer sequences.
- It appears that there may be an issue with the article's formatting, as some experiments and images are included in the "Conclusions and Future Perspectives" section. Please revise the article's layout in accordance with the journal's formatting guidelines.
The English could be improved to more clearly express the research.
Author Response
The title accurately reflects the content of the paper. The figures and tables are well-designed, and the references are of high quality. The research offers valuable insights into the mechanisms of liver fibrosis. Here are some suggestions I would like to offer:
- The interplay between RAS, RHO GTPase, and other critical signaling proteins in the activation of hepatic stellate cells (HSCs) could be described in more detail in the abstract and introduction.
Author response: Thank you very much for your time and suggestions. We have added some more details as requested in the abstract: "These findings highlight the dynamic switching between distinct pathways involving RAS, RHO GTPases, and NOTCH signaling. In particular, two specific members of RAS and RHO GTPases, ERAS and RND3, are predominantly expressed in quiescent HSCs, whereas MRAS and RHOC are more abundant in their activated forms."
In the Introduction:" Our data revealed a higher abundance of ERAS, RND3, and NOTCH1 in the quiescent state, correlating with increased activity in the ERAS-PI3K-AKT and NOTCH-RND3 pathways. Conversely, increased levels of MRAS and RHOC were associated with activated HSCs, reflecting increased activities in the RAS-MAPK and RHOC-ROCK axes."
- Please provide the animal ethics certification, and clearly describe the specific details of animal and cell procedures. The experiments in this paper seem to focus only on phenotypic validation, lacking mechanistic verification, which reduces the persuasiveness of the findings. It is recommended to include additional mechanistic experiments at the cellular level to enhance the credibility of the results. Additionally, a few schematic diagrams illustrating the proposed mechanisms could be included to provide further clarification.
Author response: The "Materials and Methods" section of the manuscript contains information that may address your concerns. "The approval number from the competent authority for animal experiments, LANUV (Landesamt für Natur, Umwelt und Verbraucherschutz Nordrhein-Westfalen, Germany; reference number 84-02.04.2015.A287), is provided to confirm compliance with animal ethics regulations."
Regarding the methods used to isolate hepatic stellate cells (HSCs), these procedures are cited in the manuscript to provide detailed insights. It's important to note that HSCs were isolated exclusively from the livers of male Wistar rats obtained from the local facility at Heinrich Heine University (Düsseldorf, Germany). No treatment was administered to the rats as their sole purpose was to isolate HSCs, thus minimizing any procedural complexity. we hope this clarification addresses your concern.
Regarding the concern about mechanistic verification, we acknowledge the reviewer's suggestion and agree that additional mechanistic insights would indeed strengthen our findings. However, at this stage, we do not have the resources or conditions to perform further mechanistic analyses. Therefore, this study focused primarily on exploring the interplay between critical signaling molecules. To substantiate the proposed mechanisms, we have provided additional supporting evidence from both previous literature and our own previous work.
In addition, the schematic diagrams illustrating the proposed mechanisms are shown in Figure 3. In the top panel of Figure 3, we have summarized the key findings of the current study, highlighting in particular the differences observed between HSCs at d0 (quiescent state) and d8 (activated state) below the arrows.
- Please clearly label the specific kDa values of the molecules in the figures. Additionally, the raw data section appears to be incomplete, as the Western blot protein markers and biological replicates are not fully provided. The authors are encouraged to improve and complete this section.
Author response: We have added clear labels indicating the specific kDa values of the molecules in all relevant figures. We have revised the raw data section to ensure that it is complete. The biological replicates and detailed information on the protein markers are now provided for clarity.
- Please ensure that the dilution ratios of all antibodies are clearly and thoroughly specified by the authors.
Author response: We have ensured that all antibody dilution ratios are listed in Table S2.
- Please clearly specify the number of base pairs in the PCR primer sequences.
Author response: The nucleotide length (number of nucleotides) for each primer sequence, both forward and reverse was added to Table S1.
- It appears that there may be an issue with the article's formatting, as some experiments and images are included in the "Conclusions and Future Perspectives" section. Please revise the article's layout in accordance with the journal's formatting guidelines.
Author response: We appreciate your feedback. We have made these changes immediately. As a result, the experiments and images have been moved to the appropriate sections.
Reviewer 2 Report
Comments and Suggestions for Authors
This investigation shows interesting evidence on the activation of HSC since authors report some proteins that might be useful as marker of HSC activation. However, some issues still need to be addressed before following up on the publication process.
- One the key phenomenon in this investigation is the activation of HSC; however, authors did not describe in material and method section the procedure about how isolated HSC were activated. It is well-known that isolated HSC get activated by them shelf in one-week once plated; however, authors must explain how they were activated.
- Since authors did not evaluate the status of all molecules shown in figure 3, the description of the proposed model, either in results section or figure legend, should be clearly explicit that the figure is a hypothetical model.
- What does stand for “d” in all figures where it appears? The meaning of “d” must be described in all figure legends.
- The number of animals (n) was not indicated anywhere in the manuscript.
- The number of cell experiments (replicated) was not indicated anywhere in the manuscript.
- In abstract section, authors wrote the following “In the affected liver, hepatic stellate cells (HSCs) transition from a quiescent to an activated state and adopt a fibrogenic phenotype similar to myofibroblast-like cells”. In this sentence, the words “similar” and “like” refer to the same description, which means that they are repetitive in the sentence. The sentence can be written as follows “…and adopt a myofibroblast-like cells phenotype”, or “…and adopt a fibrogenic phenotype similar to myofibroblasts”
- At the first appearance, authors define hepatic stellate cells and abbreviate them as HSCs; however, they cite that abbreviation as HSC; so, what is it the reason to abbreviate it as HSCs if they are not going to follow their own rules? Authors need to homogenize all abbreviations throughout the manuscript.
- When using “such as” it is It is unnecessary to write parentheses; for example, authors wrote the following “…as well as other key signaling proteins (such as NOTCH1 vs. ROCK)”. Authors should write this sentence as follows “as well as other key signaling proteins, such as NOTCH1 vs. ROCK”.
- Gene and protein short names, after definition, should be cited as their accepted symbols, and they should be indicated according to the accepted nomenclatures. For example, using combination of lowercase and uppercase letters, italics, etc., depending on the species. Throughout the manuscript including text, tables, figures, figure legends, and supplementary information, gene and protein symbols should be properly indicated. The author should know that both gene and protein short names are symbols, not abbreviations, and they are different among species. As reference, authors should review the HUGO Gene Nomenclature Committee guidelines and, GeneCards website, and read the following article: PMID: 22836666.
- For an easier validation of the evaluated gene by readers, primers shown in Supplementary Table S1 should include their respective NCBI Reference Sequence.
- The following citation in introduction section “[7][5]” must be included inside of the same bracket symbols.
- This manuscript contains some grammar issues that need to be carefully corrected by an English expert.
Please, see the comments and suggestions for authors section.
Author Response
This investigation shows interesting evidence on the activation of HSC since authors report some proteins that might be useful as marker of HSC activation. However, some issues still need to be addressed before following up on the publication process.
- One the key phenomenon in this investigation is the activation of HSC; however, authors did not describe in material and method section the procedure about how isolated HSC were activated. It is well-known that isolated HSC get activated by them shelf in one-week once plated; however, authors must explain how they were activated.
Author response: In our revised manuscript, we have provided a more detailed description of the activation process. HSCs were seeded onto rigid plastic shortly after their isolation and maintained as a monoculture in the presence of fetal calf serum (FCS). The combination of the rigid culture surface and the factors present in FCS, together with the conditions of monoculture, facilitate the activation of HSCs. Activation begins soon after cell isolation, with significant changes occurring between the 2nd and 3rd day of culture. In particular, significant changes in the transcriptome are observed following strong demethylation of DNA within the first two days after isolation. This information has been included in the Materials and Methods section [1-3].
- Since authors did not evaluate the status of all molecules shown in figure 3, the description of the proposed model, either in results section or figure legend, should be clearly explicit that the figure is a hypothetical model.
Author response: We appreciate your careful consideration. We have revised both the results section and the figure legend to clearly indicate that the proposed model shown in Figure 3 is hypothetical and based on the evaluated status of selected molecules.
- What does stand for “d” in all figures where it appears? The meaning of “d” must be described in all figure legends.
Author response: Thank you for your kind attention. We have clarified the abbreviation "d" in the figure legends. The meaning of "d" is now explicitly stated as "day" in all relevant figure legends to ensure clarity for the reader.
- The number of animals (n) was not indicated anywhere in the manuscript.
Author response: Thank you for pointing out the omission regarding the number of animals (n) used in our experiments. We have now included the number of animals in the Methods section of the manuscript. The number of animals used per experiment generally ranged from 3 to 5.
- The number of cell experiments (replicated) was not indicated anywhere in the manuscript.
Author response: We have included the number of replicates in each figure legend.
- In abstract section, authors wrote the following “In the affected liver, hepatic stellate cells (HSCs) transition from a quiescent to an activated state and adopt a fibrogenic phenotype similar to myofibroblast-like cells”. In this sentence, the words “similar” and “like” refer to the same description, which means that they are repetitive in the sentence. The sentence can be written as follows “…and adopt a myofibroblast-like cells phenotype”, or “…and adopt a fibrogenic phenotype similar to myofibroblasts”.
Author response: Thank you for your assistance in improving our manuscript. We are considering rephrasing to "...and adopt a myofibroblast-like cell phenotype" to improve clarity.
- At the first appearance, authors define hepatic stellate cells and abbreviate them as HSCs; however, they cite that abbreviation as HSC; so, what is it the reason to abbreviate it as HSCs if they are not going to follow their own rules? Authors need to homogenize all abbreviations throughout the manuscript.
Author response: Thank you for bringing this discrepancy to our attention. We have corrected the abbreviation to "HSC, Hepatic Stellate Cell". Specifically, we consistently refer to hepatic stellate cells as "HSCs" when discussing the cells in general, such as when we use HSC activation or HSC quiescence. However, we will use "HSCs" when referring to multiple cells, such as in contexts like "maintaining HSCs in a quiescent state".
- When using “such as” it is It is unnecessary to write parentheses; for example, authors wrote the following “…as well as other key signaling proteins (such as NOTCH1 vs. ROCK)”. Authors should write this sentence as follows “as well as other key signaling proteins, such as NOTCH1 vs. ROCK”.
Author response: Thanks for pointing this out! We agree that parentheses are unnecessary when using "such as" in this context. We have revised this issue.
- Gene and protein short names, after definition, should be cited as their accepted symbols, and they should be indicated according to the accepted nomenclatures. For example, using combination of lowercase and uppercase letters, italics, etc., depending on the species. Throughout the manuscript including text, tables, figures, figure legends, and supplementary information, gene and protein symbols should be properly indicated. The author should know that both gene and protein short names are symbols, not abbreviations, and they are different among species. As reference, authors should review the HUGO Gene Nomenclature Committee guidelines and, GeneCards website, and read the following article: PMID: 22836666.
Author response: Thank you for reminding us of this important issue. We have made the gene names according to the guidelines of the HUGO Gene Nomenclature Committee:: Eras, Mras, Ras, Rho, Rhoc, Notch, Rnd3, Rock, Wnt, Gfap, a-Sma, Tgf, Mapk, Akt, Yap, Mst1, Raf1, Adam17, Hippo, Hprt1, Jag1, Hes1, Col1a2, Pdk1, Arg1, Mindy1
- For an easier validation of the evaluated gene by readers, primers shown in Supplementary Table S1 should include their respective NCBI Reference Sequence.
Author response: The reference numbers are added to Table S1.
- The following citation in introduction section “[7][5]” must be included inside of the same bracket symbols.
Author response: The references are cited again and the issue is solved.
- This manuscript contains some grammar issues that need to be carefully corrected by an English expert.
Author response: Thank you for pointing out this issue. We have carefully reviewed the manuscript to ensure that the revised manuscript is free of typographical and grammatical errors.
References:
- Schumacher, E.C., et al., Combined Methylome and Transcriptome Analysis During Rat Hepatic Stellate Cell Activation. Stem Cells Dev, 2017. 26(24): p. 1759-1770.
- Götze, S., et al., Epigenetic Changes during Hepatic Stellate Cell Activation. PLoS One, 2015. 10(6): p. e0128745.
- Olsen, A.L., et al., Hepatic stellate cells require a stiff environment for myofibroblastic differentiation. Am J Physiol Gastrointest Liver Physiol, 2011. 301(1): p. G110-8.
Round 2
Reviewer 1 Report
Comments and Suggestions for Authors
The manuscript presents a compelling mechanistic study on HSC fate regulation. Key revisions include:
(1) Standardized terminology and abbreviations.
(2) Updated references and rigorous citation formatting.
Citation Style
Issue: Inconsistent citation format (e.g., “Nakhaei-Rad, 2016 #11”).
Revision: Use superscript numbers (e.g., [11]) and ensure full reference details (author names, journal abbreviations).
Updated References
Issue: Key topics (e.g., Yap in fibrosis) lack recent citations (post-2020).
Revision: Add recent reviews.
Author Response
Thank you for your careful review of our manuscript. We have checked all references and cited two additional review articles (PMIDs: 39718664 & 36471376; line 79).
Reviewer 2 Report
Comments and Suggestions for Authors
The authors have properly addressed all my recommendations.
Author Response
Thank you very much for your carefull review of our manuscript.